# PM_2.5_-Bound Polycyclic Aromatic Hydrocarbons: Sources and Health Risk during Non-Heating and Heating Periods (Tangshan, China)

**DOI:** 10.3390/ijerph17020483

**Published:** 2020-01-11

**Authors:** Bo Fang, Lei Zhang, Hao Zeng, Jiajia Liu, Ze Yang, Hongwei Wang, Qian Wang, Manman Wang

**Affiliations:** 1School of Public Health, North China University of Science and Technology, Caofeidian, Tangshan 063210, China; fangbo2018@126.com (B.F.); zhanglei_smile@126.com (L.Z.); zenghao2019@126.com (H.Z.); liujiajia913@163.com (J.L.); yangze_sunshine@163.com (Z.Y.); Wangpeople2016@163.com (H.W.); 2Hebei Province Key Laboratory of Occupational Health and Safety for Coal Industry, School of Public Health, North China University of Science and Technology, Caofeidian, Tangshan 063210, China

**Keywords:** PM_2.5_-bound PAHs, sources, incremental lifetime cancer risk, heating period

## Abstract

Exposure to polycyclic aromatic hydrocarbons (PAHs) may lead to adverse health risks. To understand the potential sources and carcinogenic risks of PAHs in Tangshan, 40 PM_2.5_ samples were collected for analysis of eighteen PM_2.5_-bound PAHs during non-heating period and heating period. The results display a significant variation. The median concentration of ∑_18_PAHs during the heating period (282 ng/m^3^) was higher than during the non-heating period (185 ng/m^3^). Especially, the median concentration of Benzopyrene (BaP) during the heating period (61.6 ng/m^3^) was 16.9-fold that during the non-heating period (3.64 ng/m^3^). It exceeded BaP annual average limit of China (1 ng/m^3^). Diagnostic ratios (DRs) and principal component analysis (PCA) both indicated that vehicle emissions and coal and biomass combustion were the dominant contributors of PAHs pollution in Tangshan. The incremental lifetime cancer risk (ILCR) of three age groups (children, teenagers, and adults) ranged from 2.56 × 10^−6^ to 5.26 × 10^−5^ during the entire sampling periods. The 95% risk values of adults exceeded 10^−4^ during the heating periods, indicating a potential health risk from PAHs.

## 1. Introduction

Particulate matter (PM) is a key indicator of air pollution, mainly including PM_10_ (particles of aero-dynamic diameter ≤ 10 μm) and PM_2.5_ (particles of aero-dynamic diameter ≤ 2.5 μm). The Global Burden of Diseases in 2015 indicates that PM_2.5_ was ranked fifth among the death risk factors, and the number of deaths attributed to ambient PM_2.5_ was 1.1 million [1]. Epidemiologic studies showed that exposure to PM_2.5_ may increase health risk of various cardiopulmonary diseases [2,3], and these adverse health effects were related to contaminants attached to PM_2.5_, such as polycyclic aromatic hydrocarbons (PAHs), polychlorinated biphenyls, and heavy metals [4].

PAHs are the primary organic component in PM_2.5_ that are formed during incomplete combustion or pyrolysis of organic matters [5]. PAHs are released from natural and anthropogenic sources [6], and it has been reported that human activities are the main source of emissions at present, including the burning of bio-organic matter, automobile exhaust emissions, and industrial emissions [7,8,9]. Increasing evidence indicates that human exposure to PAHs is positively associated with pulmonary function decline, asthma, chronic obstructive pulmonary disease, cardiovascular diseases, and cancer [10,11,12,13,14]. Considering these important health impacts, it is of great significance to analyze PAHs pollution level, possible sources, and health risks.

In Jing-Jin-Ji region, the heating periods are characterized by high coal consumption, low temperature, and inversion [15]. These factors can delay the decomposition and spread of PAHs. Studies have demonstrated that the concentrations of PM_2.5_-bound PAHs during the heating period are usually higher than during the non-heating period [16,17]. For example, in Beijing, the ratio of PM_2.5_-bound PAHs’ concentration during the heating season (72.6 ng/m^3^) to the non-heating season (4.77 ng/m^3^) is 15.2 [16]. Similarly, PM_2.5_-bound PAHs’ concentration during the heating period and the non-heating period are 288 ng/m^3^ and 101 ng/m^3^ in Tianjin, respectively [18]. Therefore, a high risk of cancer is likely to occur during the heating period. In addition, PM_2.5_-bound PAHs have a relatively high health risk and raise much attention in northern China. The 70-year lifetime risk is usually used to estimate the cancer burden of the population [19]. The United States Environmental Protection Agency’s (USEPA’s) risk assessment methods [20,21] are generally used in studies to evaluate carcinogenic and non-carcinogenic risks of PM_2.5_-bound PAHs [22,23]. However, due to the nature of multiple source signatures, pollutant-specific risk estimates may provide limited information for air quality management. Studies on incremental lifetime cancer risk (ILCR) about PM_2.5_-bound PAHs for population groups are scarce in Tangshan.

Moreover, studies have found that precipitation is an important atmospheric removal processes, which determine the residence time of pollutants in the air and the subsequent input to various ecosystems [24,25]. In this study, the sampling location was outdoors, thus it was inconvenient and would be inaccurate to sample during rainy and snowy weather, thus rainy and snowy weather were excluded. The study aimed: (1) to investigate the concentration variations of PM_2.5_-bound PAHs from October to December 2014 in Tangshan; (2) to identify the possible sources of PM_2.5_-bound PAHs; (3) to track the transfer paths and potential source regions; and (4) to assess potential carcinogenic risk attributed to PM_2.5_-bound PAHs.

## 2. Materials and Methods

### 2.1. Study Site and Sampling

According to the heating time in Tangshan, the heating period lasts from 15 November to 15 March, and the other months are grouped into the non-heating period. The study was conducted in October 2014 (non-heating period) and November to December 2014 (heating period) in Tangshan (117°31′–119°19′ E, 38°55′–40°28 N), Hebei province (Figure 1). Jing-Jin-Ji region refers to Beijing and Tianjin, and includes Tangshan as well as the seven cities of Hebei province. Tangshan is located in northern China and adjacent to Beijing and Tianjin. As a traditional coalmining city, it suffers from serious PM_2.5_ pollution. Accordingly, the PAHs exposure in PM_2.5_ is also higher, posing a high risk to human health. In recent years, due to the advantages of coal resources and geographical location, heavy industries and high energy consuming enterprises, such as steel, coal, cement, and ceramics, have developed rapidly, which poses a threat to the air quality of Tangshan as well as the entire Jing-Jin-Ji region.

We selected a sampling site of Jianshe Road on the North China University of Science and Technology campus for collection of PM_2.5_ samples. This site is close to the city center and characterized by high traffic density and commercial activities. An intelligent medium-volume air sampler (XY-2200, Qingdao Xuyu Ltd., Qingdao, China) was used to collect PM_2.5_ on quartz filters (90 mm, Whatman Inc., Maidstone, UK). The sampling was performed from 09:00 for 24 h and the flow rate was 100 L/min. Before sampling, all filters were baked at 400 °C for 12 h to remove organic impurities and PM_2.5_ cut-points were cleaned. The filters were precisely weighed before and after sampling with an electronic micro-balance (Sartorius CPA225D, 0.1 mg) right after stabilizing at 25 °C and 50% RH for 24 h. After being weighed, the filters were folded in the cartridge, marked, and stored at −20 °C until analysis.

### 2.2. Sample Preparation and Analysis

First, the filter was transferred to a 50 mL glass tube. Then, PM_2.5_-bound PAHs extraction was carried out three times using 20 mL n-hexane/dichloromethane (1:1, *v*/*v*) by ultrasound bath (25 °C, 30 min). The solvents of the three extractions were combined and concentrated by pure nitrogen. Finally, the residues were redissolved with 1 mL methanol for analysis.

The PAHs samples were quantified using a high-performance liquid chromatography with ultraviolet and fluorescence detectors (HPLC-UV-FLD). The detailed procedures were described previously [26]. The mobile phase was composed of acetonitrile and water. The ratio of acetonitrile was increased by a gradient from 60% to 100%. The detection limits of eighteen PAHs (*S*/*N* = 3) were in the range of 0.01–25 ng/m^3^ and recoveries for spiked PM_2.5_ samples ranged from 58.4% to 105.3%. Among them, the recoveries of Naphthalene (Nap), Acenaphthylene (Acy), and Acenaphthene (Ace)relatively low, approximately 60%, because of their instability.

The chemical structures of the eighteen detected PAHs are shown in Appendix A. These compounds were divided into six categories according to the number of rings: Nap (2-ring), Acy (3-ring), Ace (3-ring), Fluorene (Flu, 3-ring), Phenanthrene (Phe, 3-ring), Anthracene (Ant, 3-ring), Fluoranthene (Fln, 4-ring), Pyrene (Pyr, 4-ring), Chrysene (Chr, 4-ring), Benz[a]anthracene (BaA, 4-ring), Benz[b]fluoranthene (BbF, 5-ring), Benz[k]fluoranthene (BkF, 5-ring), Benzo[j]fluoranthene (BjF, 5-ring), BaP (5-ring), Benzo[e]pyrene (BeP, 5-ring), Indeno[1,2,3-cd]pyrene (InD, 6-ring), Dibenzo[a,h]anthracene (DbA, 6-ring), Benzo[g,h,i]-perylene (BghiP, 6-ring). PAHs can also be classified as low molecular weight (LMW-PAHs, i.e., 2- and 3-ring PAHs), medium molecular weight (MMW-PAHs, i.e., 4-ring PAHs), and high molecular weight (HMW-PAHs, i.e., 5- and 6-ring PAHs) PAHs.

### 2.3. PAHs Source Analysis

Principal component analysis (PCA) technique is widely used to identify the primary sources of PAHs [8,27]. The number of principal factors was percolated through the eigenvalue of a correlation matrix and cumulated. Values of more than 80% for the cumulative indicate and more than 1 for the eigenvalue indicator were deemed as principal components.

### 2.4. Backward Trajectory Calculation

Twenty-four-hour backward trajectories were calculated using HYSPLIT with TrajStat model for sampling periods. The starting height was set as 500 m above the ground. The GDAS dataset was obtained from NOAA’s READY Archived Meteorology online calculating program (http://ready.arl.noaa.gov/archives.php). The cluster analysis was performed to determine the transmission direction and speed of air mass. Furthermore, potential source contribution function (PSCF) analysis was also employed to analyze the possible regions of pollution source. The study region was divided into 0.5 × 0.5 of grid precision and a threshold was set (75 μg/m^3^). The trajectory can be considered as pollution trajectory when PM_2.5_ concentration corresponding to the trajectory is higher than 75 μg/m^3^. For each cell, the ratio of high points (*m_ij_*) to total points (*n_ij_*) within the cell was calculated: PSCF is defined by Equation (1), and the weight function *W*(*n_ij_*) is introduced to reduce the uncertainty of nij [28].
$${PSCF}_{ij}\;=\;\frac{m_{ij}}{n_{ij}}\;\times \;W\left(n_{ij}\right)$$
where *m_ij_* is the point number of pollution trajectories through grid (*i*, *j*) and *n_ij_* is the number of endpoints of all trajectories falling into a grid (*i*, *j*).

To reduce the uncertainties that small values of *n_ij_* produced, an empirical weight function *W_ij_* was applied as follows:(2)$$W_{ij}\;=\;\{\begin{array}{cc} 1.0 & {3n}_{ave}{<n}_{ij}\\ 0.7 & {1.5n}_{ave}<n_{ij}\leq {3n}_{ave}\\ 0.4 & n_{ave}<n_{ij}<{1.5n}_{ave}\\ 0.17 & n_{ij}\leq {3n}_{ave} \end{array}$$

### 2.5. Health Risk Assessment

The carcinogenic potential of PM_2.5_-bound PAHs was evaluated. It was based on BaP equivalent concentration (BaPeq), which was calculated by multiplying the mass concentration of individual PAHs species with their corresponding toxic equivalent factor (TEF). TEF has been widely adopted as the relative carcinogenic potency of corresponding PAHs. Different PAHs are assigned different TEF values.
(3)$${BaP}_{eq}\;=\;\overset{18}{\underset{i=1}{\sum }}{(C}_{i}{\;\times \;TEF}_{i})\;$$

In Equation (3), *C_i_* refers to the concentration of individual PAHs and *TEF_i_* means the toxic equivalent factor. The TEFs of individual PAHs are listed in Appendix A.
(4)Individual carcinogenicity index = BaPeq × cf × unit risk

In Equation (4), cf is explained as conversion factor with 0.001 µg/ng and unit risk is 1.1 × 10^−3^ (µg/m^3^)^−1^; if individual carcinogenicity index is lower than 10^−5^, public notification is not required.

ILCR model was used to quantitatively assess the potential cancer risk of the exposure to PM_2.5_-bound PAHs by inhalation pathway in the following three age groups: children (1–11), teenagers (12–17), and adults (18–70). The ILCR model for human inhalation is expressed as follows [29]:(5)$${\mathrm{ILCR}}_{i\;}{\;=(C}_{p}{\;\times \;\mathrm{IR}}_{i}{\;\times \;\mathrm{EF}\;\times \;\mathrm{ED}\;\times \;\mathrm{CSF}}_{i})/(\mathrm{BW}\;\times \;\mathrm{AT})$$

In Equation (5), Cp is the concentration of each individual PAHs (ng/m^3^), obtained by converting concentrations of PAHs to the toxic equivalents of BaP using toxic equivalency factors; *R_i_* is the incremental individual lifetime cancer risk for inhalation; EF is the exposure frequency (day/year); ED is the exposure duration (year); *CSF_i_* is the inhalation carcinogenic slope factor (mg/kg/day)^−1^; BW is body weight (kg); and AT is average time (day).

### 2.6. Sensitivity Analysis

The Monte Carlo simulation (*n* = 10,000) was implemented using Crystal Ball software to quantify the uncertainty and its impact on expected risk estimation. Incorporating variability and uncertainty into risk calculation could lead to a more realistic view of real risk allocation. Uncertainty comes from estimates of exposure and impact. Sensitivity analysis was used to determine the importance of input parameters.

## 3. Results and Discussion

### 3.1. Characteristics of PM_2.5_-Bound PAHs and Meteorology

Figure 2 shows the concentrations of PM_2.5_ and eighteen PM_2.5_-bound PAHs during non-heating and heating periods. The PM_2.5_ concentrations ranged from 23.0 to 367 µg/m^3^ with a median concentration of 93.5 µg/m^3^ during the entire period in Tangshan, which exceeds the annual average standard limits of WHO (10 µg/m^3^) and China (35 µg/m^3^). ∑_18_PAHs during the non-heating period ranged from 86.8 to 540 ng/m^3^ and during the heating period from 144 to 608 ng/m^3^. The median concentrations were 185 ng/m^3^ and 282 ng/m^3^, respectively. When *p* < 0.05, the differences of pollutant median concentrations between the heating period and the non-heating period were considered statistically significant. In present study, the results show that the concentration of PM_2.5_ was not statistically significant during the two periods, which may be related to the limited number of PM_2.5_ samples. However, the concentrations of PAHs during the heating period and non-heating period were significantly different (*p* = 0.003); the concentration of PAHs during the non-heating period was lower than during the heating period. BaP is the most toxic substance among the eighteen PAHs. The USEPA and Ministry of Environmental Protection of China recommended an annual limit value of 1 ng/m^3^ for BaP [30,31]. The median concentration of BaP during the heating period (61.6 ng/m^3^) was significantly greater than during the non-heating period (3.64 ng/m^3^) with *p* < 0.001, and the median concentration was 16.9 times higher than during the non-heating period. The temperature during the non-heating period was also found to be significantly higher than during the heating period (*p* = 0.023) in the analysis of meteorological factors (Appendix A). Therefore, it is meaningful to study the difference in PAH levels between heating and non-heating periods in Tangshan. Furthermore, PM_2.5_-bound PAH levels in our work were compared with other reports, as summarized in Table 1, implying an obvious difference in distinct regions. The PAH levels (86.8–608 ng/m^3^) in the present study were similar to those measured in Beijing (5.10–788 ng/m^3^) and Zhengzhou (7.00–961 ng/m^3^). The levels were higher than other reports in many Chinese and some non-Chinese cities, for example, Taiyuan (10.4–216 ng/m^3^), Xi’an (49.6–140 ng/m^3^), Lanzhou (2.64–352 ng/m^3^), and Inner Mongolia (0.580–180 ng/m^3^) in northern China; Lushan (1.47–25.2 ng/m^3^) and Guangzhou (8.54–122 ng/m^3^) in southern China; Atlanta (0.380–6.85 ng/m^3^) in USA; Basque (0.300–8.29 ng/m^3^) in Spain; and Tehran (2.10–410 ng/m^3^) in Iran. In northern China, a large amount of coal is consumed during the heating period in winter for domestic heating, leading to the growth of air pollutant emission [32]. Besides, the higher concentration of PM_2.5_-bound PAHs in winter may be influenced by the special weather, such as low temperature, weak wind, and lower atmospheric mixing height. On the contrary, PM_2.5_-bound PAHs’ concentration of some southern Chinese cities is relatively lower due to no heating system and unobvious seasonal differences [33].

Tangshan is located in Jing-Jin-Ji circle with serious pollution in northern China, with high energy consumption industry as its “pillar”, including coal, steel, and ceramics. According to the 2012 China statistical yearbook data, the total output of Tangshan iron and steel was 170.02 million tons, accounting for 10.8% of the country’s total steel output [34]. The PAH levels in Tangshan represent severe pollution.

The specific distribution of PAHs in different periods is shown in Figure 3. HMW-PAHs (five and six rings) were predominant, 69.59% for non-heating period and 54.13% for heating period in Tangshan. The results demonstrate that it had the most contribution to ∑_18_PAHs. Previous studies have shown that HMW-PAHs (five and six rings) are mainly related to vehicle exhaust, and MMW-PAHs (four rings) are related to coal combustion. Although the heating period in winter is linked to severe PM_2.5_ pollution, the percentage of HMW-PAHs significantly decreased from 69.58% to 54.11%. The reason may be the traffic restriction in winter, as well as closing or scaling back activity of factories. At the same time, MMW-PAHs (four rings) increased rapidly from 6.53% to 14.39% during the heating period, which may be due to the increased demand for coal.

The meteorological factors influencing on PAHs’ concentration such as temperature, relative humidity, and wind speed were analyzed using spearman correlation, as shown in Appendix A. Previous studies have shown that higher temperature would not only increase the partitioning of PAHs in the vapor phase but also cause more degradation [35]. In winter, low temperature can inhibit the degradation and transfer from particle phase into semi-volatile components of PAHs [36]. Therefore, the PAHs’ concentration in winter is relatively higher than other seasons. In this study, ∑_18_PAHs’ concentration correlated positively with temperature (*r* = −0.428, *p* < 0.001). However, there was no significant correlation between the ∑_18_PAHs’ concentration and relative humidity (*r* = −0.225, *p* > 0.05). The reason may be that the limited number of PM_2.5_ samples (*n* = 20, each period). Some studies have reported that wind speed and PAHs’ concentrations are negatively correlated [37,38], but no significant correlation between wind speed and ∑_18_PAHs’ concentration was found in this study (*r* = 0.113, *p* > 0.05).

### 3.2. Source Identification

#### 3.2.1. Diagnostic Ratios Analysis

Some PAH marker compounds and their ratios have been used as indicators to characterize and distinguish emission sources, for example petroleum, traffic related pollutants, and biomass combustion [48]. The four ratios of Ant/(Ant + Phe), Flu/(Flu + Pyr), BaA/(BaA + Chr), and InD/(InD + BghiP) were utilized for source analysis in this study, as shown in Appendix A. Ant/(Ant + Phe) < 0.1 was recommend for petroleum source, and the opposite for combustion source. Flu/(Flu + Pyr) of 0.4–0.5 indicated gasoline engine, > 0.5 biomass (such as grass and wood) and coal combustion, and <0.4 petroleum source. For BaA/(BaA + Chr), 0.35 was considered as the signal for petroleum combustion and mixed combustion of petroleum, biomass and coal. InD/(InD + BghiP) > 0.5 was related to the burning of grass, wood, and coal, a value < 0.2 was regarded as a petroleum source, and 0.2–0.5 implied petroleum combustion. Scatter plots of the four ratios during the non-heating and heating periods are presented in Figure 4. The ratio of Ant/(Ant + Phe) was 0.28 during the non-heating period and 0.43 during the heating period, indicating a combustion source (>0.1). The higher values of Flu/(Flu + Pyr) were 0.55 and 0.59 for grass, wood, and coal combustion sources. BaA/(BaA + Chr) was more than 0.35, namely 0.58 and 0.57, respectively, suggesting the dominance of petroleum combustion, such as vehicular exhaust and crude oil combustion. InD/(InD + BghiP) was 0.21 and 0.15, intimating petroleum combustion during the non-heating period and petroleum sources during the heating period. Thus, the results of ratio analysis reveal that the foremost sources of PM_2.5_-bound PAHs might be petroleum, biomass, and coal combustion during the non-heating and heating periods in Tangshan region.

#### 3.2.2. PCA Analysis

Because diagnostic ratios (DRs) were not sufficient for investigating the potential sources of PM_2.5_-bound PAHs, PCA with varimax rotation was further performed. Table 2 displays the results of PCA analysis. Three factors during the non-heating period and four factors during the heating period were extracted from the eighteen variables of PAHs.

For the non-heating period, Factor 1 (43.4% of the total variance) was mainly defined by Nap, Acy, Phe, Fln, Pyr, BaA, Chr, BbF, BkF, and BghiP. Previous studies pointed that BaA, Chr, BbF, BkF, and BghiP are associated with vehicle emissions, such as diesel and gasoline combustion. Nap, Acy, Phe, Fln, and Pyr are tracers of coal and natural gas combustion [49,50,51] Therefore, Factor 1 was interpreted as a mixture of coal combustion and vehicle emissions. Factor 2 explained 25.5% of the variance with high loadings of HMW-PAHs (including BjF, BeP, DbA, and InD). This result suggests that it may be related to gasoline-powered engine and vehicle emissions. Factor 3 (13.5% of the total variance) only consisted of Ace and Flu, which are indicators of biomass burning. For the heating period, Factor 1 (27.6% of the total variance) consisted mainly of Fln, Pyr, BaA, Chr, and BeP. Some researchers have reported that Flu, Chr, and Pyr can be explained as diesel vehicles.BaA, Fln, Chr, BeP, BbF, and BkF are considered as indicators of coal combustion and stationary source from factories. Fln indicates fossil fuel combustion and industrial process, such as coke production [52,53]. Factor 2 (including NaP, Ace, Flu, and BjF) and Factor 4 (only Acy) explained 24.3% and 10.2%, respectively. Nap, Ace, Acy, and Flu are indicators for coal combustion and BjF for vehicle emissions [54]. Factor 3 (Ant, BkF, and BghiP) explained 25.5% of the variance. BkF and BghiP are a result for vehicle exhaust, especially BghiP is strongly associated with traffic origin, mainly gasoline and diesel emissions. Ant indicates coal combustion [55]. To improve air quality, Tangshan municipal government has taken measures such as traffic control and restricting enterprise production. Therefore, PAHs mainly derived from motor vehicles may be lower during the heating period than during the non-heating period.

Overall, PCA results indicate that the vehicle emissions and coal and biomass combustion were the dominating sources during the non-heating period and heating period, which is consistent with DRs. This suggests that traffic and coal are the main sources of PAHs in Tangshan. As a typical heavy industry city, vehicle exhaust pollution has possibly become one of the main causes due to unreasonable transportation structure. In addition, the execution of policies of coal-to-gas system should be further accelerated to reduce the contribution of coal combustion during the heating period in Tangshan.

#### 3.2.3. Backward Trajectory Analysis

To understand the effect of atmospheric transmission to PAHs source, the HYSPLIT with TrajStat model was performed to calculate the trajectories of air mass after 24 h. Figure 5 presents the results of cluster analysis and weight PSCF (WPSCF).

During the non-heating period (Figure 5a), the air trajectories from northwest (NW, Clusters 1, 3, and 4) account for 63.31% of the total trajectories, and transfer speed is faster than southwest (SW, Cluster 2, 36.69%). The transfer paths of NW originated in Mongolia and Inner Mongolia, passing through Jing-Jin-Ji region to Tangshan. Another path originated mainly at the boundary of Shandong province and Hebei province with a relatively short transmission distance. As shown in Figure 5c, these transfer directions of Clusters 2–4 (NW) and 1 (SW) during the heating period were similar to NW (Clusters 2–4) and SW (Cluster 1) of the non-heating period. In addition, Cluster 5 (24.73%) showed that there was a circle of local pollutant trajectories. Furthermore, PM_2.5_ average concentration was calculated by cluster statistics. PM_2.5_ average concentration of trajectories from NW and SW during the non-heating period were 42.5 μg/m^3^ and 36.4 μg/m^3^, respectively. The pollution was obviously aggravated when entering the heating period; PM_2.5_ average concentration of trajectories from NW was 146 μg/m^3^, trajectories of SW for 195 μg/m^3^, and trajectories of local for 128 μg/m^3^. From the results of WPSCF, the range of potential source regions in heating were significantly wider than non-heating. The greater value of WPSCF in Tangshan was mainly concentrated in Mongolia, south of Hebei province, and west of Shandong province (Figure 5b). It showed that these areas were the main potential source regions of PM_2.5_-bound PAHs during the non-heating period. During the heating period, due to coal combustion and unfavorable meteorological factors, the area of potential source regions expanded. It was mainly concentrated in Mongolia, Inner Mongolia, Jing-Jin-Ji, and west of Shandong province (Figure 5d).

In summary, the pollution of PM_2.5_-bound PAHs during the sampling periods in Tangshan was mainly affected by local pollution sources and long-distance transmission. The potential source regions of PM_2.5_-bound PAHs were possibly Mongolia, Inner Mongolia, Jing-Jin-Ji, and Shandong province, as well as locally in Tangshan. Moreover, the contribution rate of the air mass from NW to PM_2.5_-bound PAHs was significantly higher than SW, which may be related to coal combustion in winter. At the same time, the local pollution contribution of Tangshan cannot be ignored, which may be attributed to industrial activities and coal combustion.

### 3.3. Health Risks Assessment of PM_2.5_-Bound PAHs

From the results in Appendix A, it is clear that BaP concentrations had significant variation. The BaP concentration during the heating period (61.6 ng/m^3^) was 16.9 foldthat during the non-heating period (3.64 ng/m^3^). The total BaPeq of each PAHs was calculated to appraise the carcinogenic efficacy. The result for total BaPeq also shows that the heating period (71.4 ng/m^3^) had a greater concentration than the non-heating period (27.2 ng/m^3^). The range of 1.0 × 10^−6^–1.0 × 10^−5^ was within the acceptable individual cancer risk values according to the recommendations of USEPA and WHO. If the value of individual cancer risk exceeds 1.0 × 10^−5^, relevant measures should be taken to reduce the risk to the population, such as single and double number limit of vehicles and limit production of factories. The individual carcinogenicity indices of PAHs were different (heating period > non-heating period) in our study. Individual carcinogenicity indices during the heating period (7.85 × 10^−5^) and non-heating period (2.99 × 10^−5^) demonstrated that PAHs may be a risk for cancer.

Considering the sensitivity of age, three groups of children, teenagers, and adults were included in the ILCR model for risk assessment. The risk parameters and the range of ILCR values in different age groups are shown in Table 3. The value of ILCR from 10^−6^ to 10^−4^ indicated a potential risk according to the USEPA [56] ILCR ≥ 10^−4^ is deemed a serious risk, which means that more attention should be paid to such health issues. In the present study, ILCR results suggest the order adults > teenagers > children during entire period. The reason might be that adults have more exposure to pollutants compared with children. The carcinogenic risk of PAHs by inhalation was also investigated; children were exposed to PAHs mainly by intake and dermally. Consequently, ILCR was still limited to assessing health risks by considering only inhalation exposure. This results of ILCR might underestimate the total risk (including inhalation, intake, and dermal exposure).

Table 4 lists the probabilistic risk values for all age groups as a percentage of the value during the non-heating and heating periods. As can be noted, the risk values for each percentile are both in the order children < teenagers < adults during the non-heating and heating periods, and all age groups had low risk values (10^−6^ < ILCR < 10^−4^) during the non-heating periods, which indicates potential risk. Similarly, the health risk value of children and teenagers is lower than the risk value. However, the 95% risk values of adults exceeded 10^−4^ during the heating periods. Adult ILCR values were analyzed. Special attention should be paid to this group as most people within the sampling area are aged 18–70. Source code control is the most effective and important measure to be taken. Therefore, Tangshan municipal government should take effective measures to reduce PAHs’ concentration.

### 3.4. Sensitivity Analysis

This study performed a quantitative sensitivity analysis to assess the variability and uncertainty of parameters in the exposure pathway that have the greatest impact on the risk estimates, as shown in Figure 6. The sensitivity analysis results of the inhalation risk model were shown in the form of tornado chart, which illustrates the spearman rank order correlation coefficient. For inhalation exposure to PM_2.5_-bound PAHs, CSF_i_ is the most influential variable during the non-heating and heating periods, which contributions to output variance range of 30.7–90.0% and 31.1–91.3%, respectively. The sensitivity analysis showed that efforts should be made to better define the probability distribution of CSFi to improve the accuracy of results.

### 3.5. Limitations

The limitations of this study lie in the short sampling period (autumn and early winter season) of PM_2.5_. Due to the high cost of PAHs measurement and weather conditions, only one sampling point was set up in the study, and ILRC only considered the health risks exposed by respiratory route, which may underestimate the total risks (including intake and dermal exposure). Therefore, in future research, more representative sites and more PAHs measurements should be selected to better show the regional public health risk status.

## 4. Conclusions

From October to December 2014, the average mass concentration of PM_2.5_ was 122 µg/m^3^, with a range from 23.0 to 367 µg/m^3^. Sixty-three percent of the measured values exceeded China’s national standard (75 µg/m^3^, 24 h) in Tangshan. The median concentrations of PM_2.5_-bound PAHs’ concentrations were 227 ng/m^3^, with a range from 86.8 to 608 ng/m^3^ during the sampling period. The median concentrations of PM_2.5_-bound PAHs was 185 ng/m^3^ during the non-heating period and 282 ng/m^3^ during the heating period. It should be noted that median concentration of BaP during the heating period (61.6 ng/m^3^) was significantly higher than during the non-heating period (3.64 ng/m^3^). The levels exceeded the BaP annual average limit of China (1 ng/m^3^). Based on the DRs and PCA results, vehicle emissions and coal and biomass combustion are the major contributors to PM_2.5_-bound PAHs in Tangshan. Furthermore, the results of backward trajectory indicate that the pollution of PM_2.5_-bound PAHs during the sampling periods in Tangshan were mainly influenced by air masses from NW and SW and local pollution sources. The potential source regions of PM_2.5_-bound PAHs were Mongolia, Inner Mongolia, Jing-Jin-Ji, and Shandong province, as well as locally in Tangshan. The ILCR value of different age groups during the sampling periods exceeded the acceptable level (10^−6^); the 95% risk values of adults exceeded 10^−4^ during the heating periods. which suggests that PAHs might be a potential carcinogenic risk to human health in Tangshan.

## Figures and Tables

**Figure 1 ijerph-17-00483-f001:**
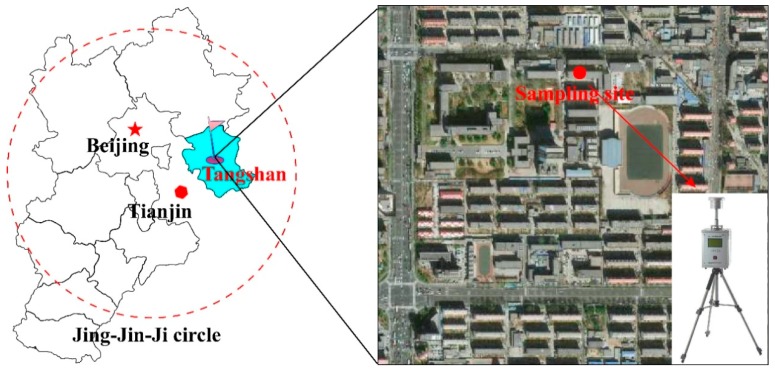
The location of sampling site in Tangshan, China.

**Figure 2 ijerph-17-00483-f002:**
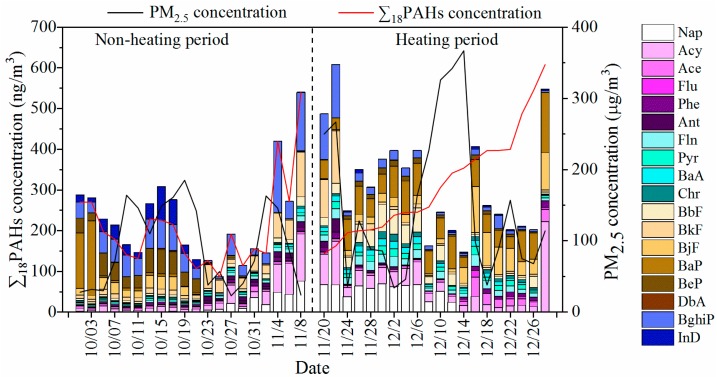
The concentrations of PM_2.5_ and eighteen PAHs in PM_2.5_ from October to December 2014.

**Figure 3 ijerph-17-00483-f003:**
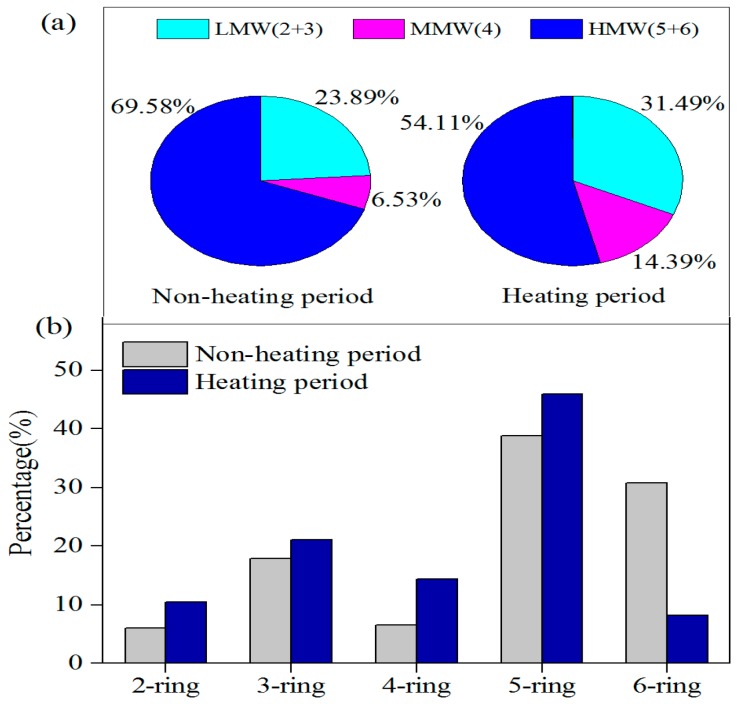
Distributions of (**a**) ring number and (**b**) molecular weight of PM_2.5_-bound PAHs during the non-heating and heating periods.

**Figure 4 ijerph-17-00483-f004:**
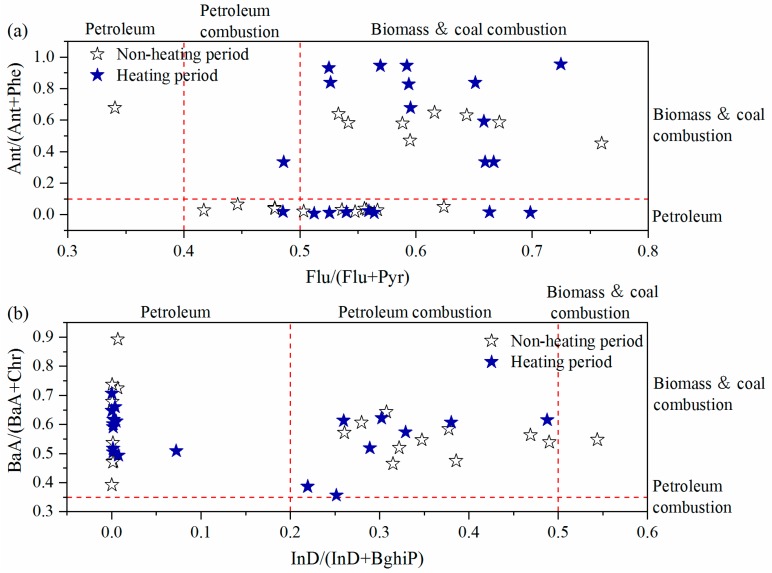
The results of diagnostic ratios for: (**a**) Flu/(Flu + Pyr) vs. Ant/(Ant + Phe); and (**b**) InD/(InD + BghiP) vs. BaA/(BaA + Chr).

**Figure 5 ijerph-17-00483-f005:**
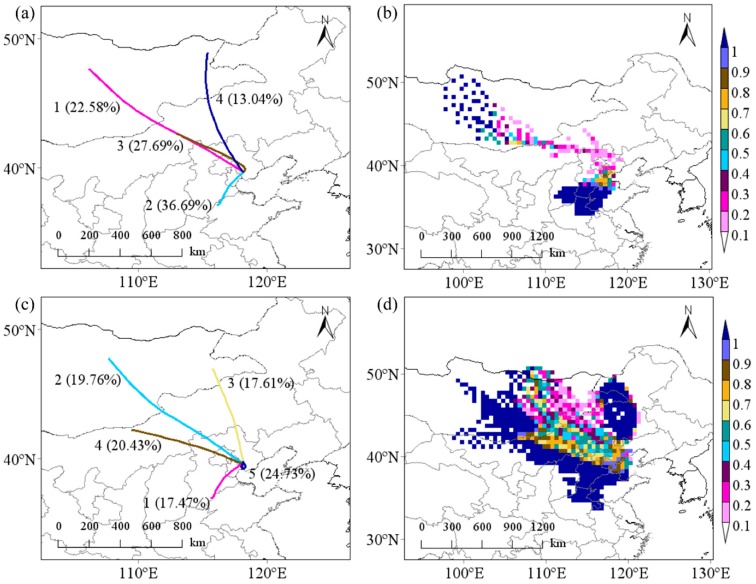
The cluster analysis and WPSCF of backward trajectories in Tangshan during: (**a**,**b**) the non-heating period; and (**c**,**d**) the heating period.

**Figure 6 ijerph-17-00483-f006:**
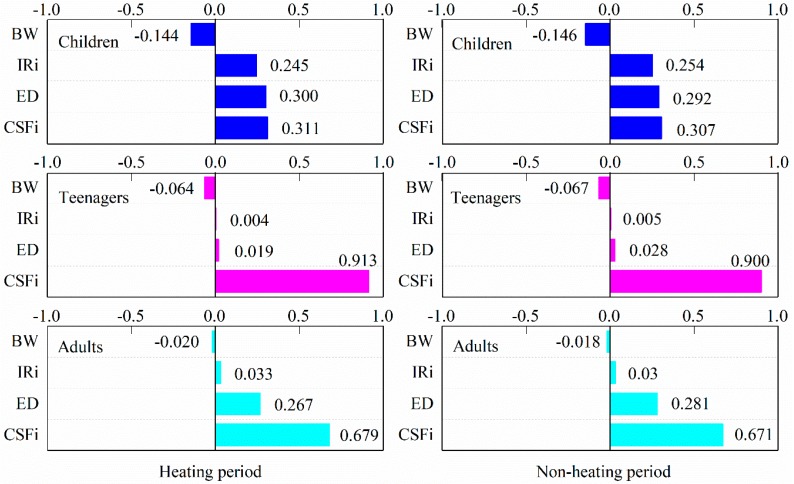
Sensitivity analysis for inhalation ILCR model for three age groups during the non-heating and heating periods.

**Table 1 ijerph-17-00483-t001:** Comparisons with PM_2.5_ (μg/m^3^) and PM_2.5_-bound PAHs (ng/m^3^) values derived from previous studies.

County	City	Periods	PM_2.5_	∑PAHs	Type of Location	References
China	Tangshan	2014	23.0–367	86.8–608	Urban	This work
China	Beijing	2014–2016	13.6–266	5.10–788	Urban	[39]
China	Zhengzhou	2011–2013	55.0–697	7.00–961	Suburb	[40]
China	Lushan	2012	63.9–428	1.47–25.2	Mount	[41]
China	Guangzhou	2007–2008	9.37–148	8.54–122	Urban	[42]
China	Taiyuan	2012	- ^a^	10.4–216	University	[29]
China	Xi’an	2012	- ^a^	49.6–140	Middle-school	[43]
China	Inner Mongolia	2005	- ^a^	0.580–180	Urban	[44]
China	Taiwan	2014–2015	9.40–88.6	1.33–6.04	Urban	[45]
USA	Atlanta	2014	- ^a^	0.380–6.85	Urban	[46]
Spain	Basque	2006–2011	- ^a^	0.300–8.29	Urban	[47]

^a^ not available.

**Table 2 ijerph-17-00483-t002:** Factor loadings of PCA analysis for eighteen PAHs during the non-heating and heating periods.

PAHs	Non-Heating Period			Heating Period			
	Factor 1	Factor 2	Factor 3	Factor 1	Factor 2	Factor 3	Factor 4
Nap	0.854				−0.718		
Acy	0.905						0.768
Ace			0.969		0.964		
Flu			0.944		0.783		
Phe	0.865						
Ant						0.932	
Fln	0.854			0.899			
Pyr	0.835			0.875			
BaA	0.745			0.936			
Chr	0.910			0.909			
BbF	0.846						
BkF	0.816					0.970	
BjF		0.930			0.924		
BaP							
BeP		0.915		0.852			
DbA		0.845					
BghiP	0.799					0.944	
InD		0.959					
Total variance (%)	43.4	25.5	13.5	27.6	24.3	22.2	10.2
Sources	Coal combustionand and vehicle	Vehicle	Biomass burning	Coal, fossil fuel and vehicle	Coal combustion	Coal combustion and vehicle	Coal combustion

**Table 3 ijerph-17-00483-t003:** Risk parameters and the range of ILCR values in different age groups by inhalation.

Parameters	Children (1–11)		Teenagers (12–17)		Adults (18–70)	
	Male	Female	Male	Female	Male	Female
BW ^a^	(17.2, 6.30)	(16.5, 6.20)	(47.1, 9.80)	(44.8, 7.40)	(60.2, 2.90)	(53.1, 2.80)
IRi ^b^	(14.1, 1.72)		(32.1, 1.04)		(32.7, 1.14)	
EF ^c^	(252, 1.01)		(252, 1.01)		(252, 1.01)	
AT ^d^	25,550		25,550		25,550	
ED ^e^	1–11		12–17		18–70	
CSFi	3.14		3.14		3.14	
Non-heating period	0.430 × 10^−6^– 4.74 × 10^−6^	0.440 × 10^−6^– 4.87 × 10^−6^	6.02 × 10^−6^– 8.53 × 10^−6^	6.22 × 10^−6^– 8.82 × 10^−6^	7.81 × 10^−6^– 30.4 × 10^−6^	8.20 × 10^−6^– 31.9 × 10^−6^
Heating period	1.13 × 10^−6^– 12.4 × 10^−6^	1.16 × 10^−6^– 12.8 × 10^−6^	15.8 × 10^−6^– 22.4 × 10^−6^	16.3 × 10^−6^– 23.1 × 10^−6^	20.5 × 10^−6^– 79.7 × 10^−6^	21.5 × 10^−6^– 83.6 × 10^−6^

BW, body weight (kg); IR_i_, the incremental individual lifetime cancer risk for inhalation; EF, the exposure frequency (day/year); AT, average time (day); ED, the exposure duration (year); CSF_i_, the inhalation carcinogenic slope factor (mg/kg/day)^−1^. ^a^ Adapted from Department of Health, ROC (http://www.doh.gov.tw/cht/index.aspx#); ^b^ adapted from ICRP 66 (ICRP, 1994)**;**
^c^ adapted from Central Personnel Administration, ROC; ^d^ adapted from USEPA (2001); ^e^ adapted from US Environmental Protection Agency, Washington (2011).

**Table 4 ijerph-17-00483-t004:** Probabilistic carcinogenic risk values of PAHs for all age groups during non-heating and heating Periods.

Seasons	Percentile	Children	Teenagers	Adults
Non-heating Period	5%	5.81 × 10^−7^	3.01 × 10^−6^	5.93 × 10^−6^
	25%	1.76 × 10^−6^	5.57 × 10^−6^	1.22 × 10^−5^
	50%	3.78 × 10^−6^	8.34 × 10^−6^	2.02 × 10^−5^
	75%	7.87 × 10^−6^	1.28 × 10^−5^	3.26 × 10^−5^
	95%	1.96 × 10^−5^	2.31 × 10^−5^	6.31 × 10^−5^
Heating Period	5%	1.58 × 10^−6^	7.81 × 10^−6^	1.58 × 10^−5^
	25%	4.74 × 10^−6^	1.44 × 10^−5^	3.22 × 10^−5^
	50%	1.00 × 10^−5^	2.19 × 10^−5^	5.27 × 10^−5^
	75%	2.04 × 10^−5^	3.30 × 10^−5^	8.56 × 10^−5^
	95%	5.36 × 10^−5^	6.12 × 10^−5^	1.70 × 10^−4^

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
