# Peer review of "PM2.5-Bound Polycyclic Aromatic Hydrocarbons: Sources and Health Risk during Non-Heating and Heating Periods (Tangshan, China)"

_ijerph, 2020, doi:10.3390/ijerph17020483_

Round 1

Reviewer 1 Report

This paper deals with the comparison of the presence of particle bonded PAH in two seasons. Diagnostic ratio analysis and Principal Component Analysis was used for source determination and Hysplit to obtain backward trajectories.

The study is important for all regions since PAH are important mutagenic pollutants, nevertheless I have some concern about the selection of seasons since I think that 20 days of sampling are not enough to define two seasons, when the months are together; in addition the risk analysis is incomplete. Then, more work should be done before publication.

In the first place, authors should explain why October is a quite different season (no heating) that November and December (heating). How was made the selection. The differences between the seasons should be explained in a Table showing: temperatures, thermal inversions, wind speed and directions, since in the papers they mentioned as references the studies were made for one or two complete year where there is a clear separation of seasons. The Table S2 clearly shows that there is not significant difference in PM2.5. It should be clarify why October is not in the heating season. PAH concentrations of the last two days during “no heating” period are quite similar than those of the two first days of “heating” period”, then I think that the selection of dates is correct, authors must convince the readers that indeed there are two seasons; at least authors have to do a better discussion. As I said before 20 days are few days to define different seasons. Maybe it would be better no to talk about seasons but study the atmospheric and weather conditions in the different sampled days.

In the other hand, the risk analysis carried out is very simple, the authors only made the multiplications recommended to obtain ILCR but I am afraid that the incremental lifetime cancer risk (ILCR) performed in this study is not valid since the uncertainty analysis is missing, and the USEPA method is incomplete. Then, for a valid health risk analysis a Montecarlo simulation should be carried out as recommended for Zhang et al 2016,  that is the reference that authors cited in the manuscript, but they did not follow it in the present study. If not, the term health risk analysis should be eliminated from the tittle.

The diagnostic ratios are almost the same in both seasons as well as PCA results. These confirm that maybe all data should be treated and discussed together, showing the high concentrations and sources of PAH. BAPeq of BkF  is higher during no heating period, showing again that there are not two seasons.

Reviewer 2 Report

General Comment:

The paper entitled: “PM2.5-bound Polycyclic Aromatic Hydrocarbons: Sources and Health Risk in Non-heating and Heating Periods (Tangshan, China)”, by authors Bo Fang et al., was well written and conveyed an important though bleak air quality and human health condition in the industrialized city of Tangshan. It is a scientific whistle blower.

I commend the authors for choosing a rather acute situation of public health risk in Tangshan, China. The broad stroke brush description was bleak and urgent. The authors also based their attribution studies on a vast collection of relevant measurement studies in the same geographical area and similar meteorological conditions. Therefore, the basis of good quality data extent to roughly a large part of northeastern China and for the fall and winter seasons.
The fall and early winter season chosen in 2014 however is rather limited in the data sample to substantiate a conclusive ascertainment of where and how-much the attribution of the PAHs were. Having said that I understand the challenge to collect and analyze a large data set at multiple monitoring sites in Tangshan --- by itself is a rather geographically large and highly variable topological, emission-driven, and meteorological winter conditions. The authors did a good skillful effort to complement the versatility of their data set with good measurements of PAHs in a large region (e.g., Shenyang (Yang et al., 2019)).
Deposition processes are key factors determining the surface concentrations of primary pollutant especially for aerosol species. The majority of the suspended particulate species near surface are emitted likely as primary emissions – such as fugitive dust and pyro-particles from biomass burning and/or anthropogenic sources. The surface characteristics are governing factor on depositional velocity as well as on the likelihood of re-suspension. The authors are right in choosing the winter condition as the interest of season for PM2.5 studies as it is the most polluted period due to stagnation and poor meteorological ventilation. However, the largest wholescale characteristic change of the surface in winter is snowfall events. It change the depositions significantly both during precipitation with washout/accretion/scavenging and afterwards with snow-cover that strongly modifies dry depositional speeds. I would encourage the authors to mention this in the introductory section to warn the readers that this study did not consider snow-cover conditions. It is a big study by itself therefore the authors can cite some literature and tag that as a possible future work.

Detailed comment:
Typo in line 234:
..(43.4% of the total variance) was led by Nap, Acy ….. (I do not think the authors meant leaded as “putting an additive in gasoline”)

Reviewer 3 Report

The paper deals with the determination of PM2.5-bound Polycyclic Aromatic Hydrocarbons Sources and Health Risk during Non-heating and Heating Periods in China. The paper deals with a very interesting topic that fits with the aims of this journal. In the following comments and suggestions to improve the paper are reported. Other comments have been indicated directly in the text.

 The information on the sampling site from line 43 to line should be moved in material and methods section.A huge literature is present dealing with the contribution of different sources to PAH emission. Therefore I suggest thata t line 39 together with reference 7 the following references should be included:

Piazzalunga, A., Anzano, M., Collina, E., Lasagni, M., Lollobrigida, F., Pannocchia, A., Fermo, P., Pitea, D. Contribution of wood combustion to PAH and PCDD/F concentrations in two urban sites in Northern Italy (2013) Journal of Aerosol Science, 56, pp. 30-40) and in particular from agricultural residues burning such as that mentioned by the authors (puddy-residue burning). This reference could be included: Vassura, I., Venturini, E., Marchetti, S., Piazzalunga, A., Bernardi, E., Fermo, P., Passarini, F. Markers and influence of open biomass burning on atmospheric particulate size and composition during a major bonfire event (2014) Atmospheric Environment, 82, pp. 218-225. 

 At line 105 include again as reference Piazzalunga et al. 2013 .

Some references should be added at line 209 and further on as regards as the diagnostic ratios; among these a paper where these ratios are considered and applied is:

Fasani, D., Fermo, P., Barroso, P.J., Martín, J., Santos, J.L., Aparicio, I., Alonso, E.; Analytical Method for Biomonitoring of PAH Using Leaves of Bitter Orange Trees (Citrus aurantium): a Case Study in South Spain (2016) Water, Air, and Soil Pollution, 227 (10), art. no. 360; DOI: 10.1007/s11270-016-3056-z

The paragraph dealing with the assessment of health risk should be improved clarifying better. For example not all the parameters reported in table 3 have been clearly described so please improve this part.

Round 2

Reviewer 1 Report

The authors adequately responded to all recommendations and the document substantially improved